# Efficient Selection of Antibodies Reactive to Homologous Epitopes on Human and Mouse Hepatocyte Growth Factors by Next-Generation Sequencing-Based Analysis of the B Cell Repertoire

**DOI:** 10.3390/ijms20020417

**Published:** 2019-01-18

**Authors:** Soohyun Kim, Hyunho Lee, Jinsung Noh, Yonghee Lee, Haejun Han, Duck Kyun Yoo, Hyori Kim, Sunghoon Kwon, Junho Chung

**Affiliations:** 1Department of Biochemistry and Molecular Biology, Seoul National University College of Medicine, Seoul 00380, Korea; kimchii481@gmail.com (S.K.); dk93js@gmail.com (D.K.Y.); 2Cancer Research Institute, Seoul National University College of Medicine, Seoul 00380, Korea; 3Department of Electrical Engineering and Computer Science, Seoul National University, Seoul 08826, Korea; eyebis208@gmail.com (H.L.); temporaljoys@gmail.com (J.N.); yonghee9141@gmail.com (Y.L.); 4Celemics, Inc., 131 Gasandigital 1-ro, Geumcheon-gu, Seoul 08506, Korea; hhjunny@gmail.com; 5Department of Biomedical Science, Seoul National University College of Medicine, Seoul 00380, Korea; 6Genomic Medicine Institute (GMI), Medical Research Center, Seoul National University, Seoul 00380, Korea; 7Convergence medicine research center, Asan Institute for Life Sciences, Asan Medical Center, Seoul 05505, Korea; hyol409@gmail.com; 8Department of Convergence Medicine, University of Ulsan College of Medicine, Seoul 05505, Korea; 9Interdisciplinary Program in Bioengineering, Seoul National University, Seoul 08826, Korea; 10Institutes of Entrepreneurial BioConvergence, Seoul National University, Seoul 08826, Korea; 11Seoul National University Hospital Biomedical Research Institute, Seoul National University Hospital, Seoul 03080, Korea; 12Inter-University Semiconductor Research Center, Seoul National University, Seoul 08826, Korea

**Keywords:** next-generation sequencing, hepatocyte growth factor, B cell receptor, immune profiling

## Abstract

YYB-101 is a humanized rabbit anti-human hepatocyte growth factor (HGF)-neutralizing antibody currently in clinical trial. To test the effect of HGF neutralization with antibody on anti-cancer T cell immunity, we generated surrogate antibodies that are reactive to the mouse homologue of the epitope targeted by YYB-101. First, we immunized a chicken with human HGF and monitored changes in the B cell repertoire by next-generation sequencing (NGS). We then extracted the *V_H_* gene repertoire from the NGS data, clustered it into components by sequence homology, and classified the components by the change in the number of unique *V_H_* sequences and the frequencies of the *V_H_* sequences within each component following immunization. Those changes should accompany the preferential proliferation and somatic hypermutation or gene conversion of B cells encoding HGF-reactive antibodies. One component showed significant increases in the number and frequencies of unique *V_H_* sequences and harbored genes encoding antibodies that were reactive to human HGF and competitive with YYB-101 for HGF binding. Some of the antibodies also reacted to mouse HGF. The selected *V_H_* sequences shared 98.3% identity and 98.9% amino acid similarity. It is therefore likely that the antibodies encoded by them all react to the epitope targeted by YYB-101.

## 1. Introduction

Hepatocyte growth factor (HGF), also known as scatter factor, is a ligand for c-MET and was initially identified as a growth factor for fibroblast-derived cell motility factor and hepatocytes [1]. Secreted by fibroblasts and mesenchymal cells, and also by epithelial cells under special circumstances [2], HGF is produced in an inactive state and is converted by proteolysis into its active heterodimeric form consisting of a 69 kDa α-chain subunit and a 34 kDa β-chain linked by a disulfide bond [3]. The active form consists of an amino (N) domain, four Kringle domains (K1–K4) in the α-chain, and a serine proteases homology domain (SPH) in the β-chain [4,5]. The binding of active HGF to c-MET activates signaling cascades leading to cancer progression, invasion, and metastasis [6]. Currently, there are four anti-HGF antibodies in clinical trials, including rilotumumab (AMG-102), ficlatuzumab (AV-299), HuL2G7 (TAK-701), and YYB-101 [7,8,9,10]. 

Antibodies that are cross-reactive to mouse HGF are needed to test the in vivo effects of antibody-mediated HGF neutralization. Despite 90.3% identity and 95.6% similarity between human and mouse HGF, YYB-101 and the other antibodies in clinical trials are not reactive to mouse HGF. There has been no report of a mouse HGF-neutralizing monoclonal antibody. Therefore, a knock-in mouse with human HGF in an immunodeficient NOD scid gamma (NSG) background was generated and used for in vivo study [11]. 

Recently, it was shown that neutrophils recruited to T cell-inflamed microenvironments rapidly acquired immunosuppressive properties [12]. The inhibition of HGF/c-MET signaling impaired those acquired immunosuppressive properties and also reduced the exhaustion of cytotoxic T cells [13]. In patients with cancer, high serum levels of HGF were correlated with increasing neutrophil counts and unresponsiveness to anti-PD-1 checkpoint blockade [12]. All of those observations suggested that treatment with HGF-neutralizing antibodies might potentiate the efficacy of immune checkpoint inhibitors. That hypothesis cannot be tested in human-HGF knock-in NSG mice, however, because NSG mice lack T cells. The most accurate way to test the combinational therapeutic effects of YYB-101 would be to treat immunologically intact mice with a mouse HGF-neutralizing antibody that binds to the homologous epitope on human HGF. 

To obtain an antibody with those characteristics, we immunized a chicken with human HGF, monitored the chronological change in the B cell receptor repertoire using next-generation sequencing (NGS), and analyzed the change in the B cell repertoire using an algorithm developed in this study. We identified groups of variable heavy-chain (*V_H_*) transcripts that underwent significant change following the immunization. We generated a phage-display single-chain variable fragment (scFv) library and selected reactive clones in a high-throughput manner as described previously [14]. The reactive clones retrieved from the phage-display library contained *V_H_* genes that had changed significantly following the immunization. From those reactive clones, we could successfully select antibodies that were reactive to both human and mouse HGF and competitive with YYB-101 for binding to human HGF. 

## 2. Results

We immunized one chicken with human HGF and boosted the immunization twice at the second week and the fourth week, as shown in Table 1. Peripheral blood was collected before the immunization (week 0), at the times of the first and second boosters (weeks 2 and 4), and one week after the second booster (week 5), just before sacrifice. After isolation of mononuclear cell fraction from the blood, RNA was prepared to produce cDNA. We used the cDNA and specific primers to amplify the *V_H_* gene, which we sequenced on the Illumina MiSeq NGS platform. NGS data was preprocessed by quality-based filtering and error correction based on hierarchical clustering as described previously [15]. Thus, we obtained 133,312 unique *V_H_* nucleotide sequences from the four *V_H_* transcript sets corresponding to each of the blood samples, respectively. 

To explore these 133,312 *V_H_* nucleotide sequences, we used network analysis tools described previously [16,17,18]. Each individual *V_H_* transcript and its read count were represented as a vertex and its size, respectively. Levenshtein distance (LD), defined as the number of mutations between two *V_H_* transcripts, was employed to reflect the degree of similarity between each and every pair of vertices [19]. We then grouped the vertices by applying “edges” to connect pairs for which the LD was below an artificially defined threshold. For example, when the LD threshold was set as two, vertices with an LD of one or two between them were connected by an edge. We refer to the resulting groups of vertices connected by edges as “components,” as described previously [20]. It was critical to find the optimal LD threshold to properly compartmentalize related *V_H_* transcripts into components while minimizing the inclusion of unrelated *V_H_* transcripts. If the LD threshold was set too low, *V_H_* transcripts with high numbers of somatic mutations would fail to be incorporated into the proper component. If the threshold was set too high, too many unrelated *V_H_* transcripts would be included in the components. To determine the optimal LD threshold, we focused on the complementary determining region 3 (CDR3) sequence. HCDR3 (CDR3 of *V_H_*) plays a critical role in antigen-antibody interactions and has a relatively low rate of replacement mutation [21]. We defined the HCDR3 ratio as the number of unique HCDR3 amino acid sequences divided by the number of vertices in each component, as shown in Figure 1a. We found that when we set the LD threshold between 4 and 15, a large number of vertices had an HCDR3 ratio between 0.1 and 0.3, while few had an HDCR3 ratio between 0.3 and 0.4. From that observation, we hypothesized that there are discrete fractions of components with HCDR3 ratios between 0.1 and 0.3. To include the maximum number of vertices in our subsequent analysis, we used an LD threshold of 12, which gave the maximal number of vertices in components with an HCDR3 ratio less than 0.3, as shown in Figure 1b. As a result, the dataset of 133,312 vertices was reconstituted into 104,992 components connected by 944,649 edges. Among those, 99,084 of the components were composed of a single vertex. We excluded the components with a single vertex that did not contain any edges, which left 5908 components for further analysis. 

To select components containing *V_H_* transcripts that encoded human HGF-reactive antibodies, we focused on the changes in antigen-reactive B cells after immunization with HGF. B cells with human HGF-reactive receptors will produce daughter cells with higher-affinity receptors via somatic hypermutation and/or gene conversion [22]. Therefore, components that contain *V_H_* transcripts for HGF-reactive B cell receptors (BCRs) should show an increase in the number of vertices that they contain over time. The network of all vertices in four sets with edges determined by the minimum spanning tree method, as shown in Figure 2a [23], contained components composed of vertices that were newly formed as a result of the immunization and boosting, as shown in Figure 2 and Appendix A. We expected that preferential proliferation of B cells with HGF-reactive BCRs would cause the read counts for vertices representing those BCRs to increase. We defined the clonal frequency as the read count of a given vertex divided by the total read count of the *V_H_* transcript set in the given NGS analysis. We defined the average clonal frequency as the average value of the clonal frequency of all the vertices within a given component. For the naming of the components, we sorted all the components with the number of vertices throughout four time sets in descending order and gave them an identification number in an ascending order. We found that certain components—including components 3, 20, and 37—showed a significant increase in the number of vertices and also in the average clonal frequency from week 0 to week 5, as shown in Figure 2b, Appendix A.

To validate the specific binding of the antibodies encoded by *V_H_* transcripts to human HGF, we constructed a phage-display scFv library with a complexity of 1.1 × 10^9^ using the cDNA of the mononuclear cell fraction obtained at week 5. The scFv clones were retrieved after the fourth round of bio-panning on human HGF in a high throughput manner as described previously [14]. We rescued a total of 306 clones and subjected them to phage enzyme-linked immunosorbent assay (ELISA). In the phage ELISA, a total of 218 clones reacted to human HGF. Within components 3, 20, and 37 showing a significant increase both in the number of vertices and in the average clonal frequency, 1 out of 31, 193 out of 195, and 1 out of 5 ELISA-subjected clones reacted to human HGF, respectively, as shown in Appendix A. Component 20 harbored the highest number of positive clones, as shown in Figure 3a,b and Appendix A. The *V_H_* sequences in component 20 shared 98.3% identity and 98.9% amino acid similarity. We tested the reactivity of the clones to mouse HGF and found that many of the clones in component 20 bound to mouse HGF, as shown in Figure 3b. Clones with various binding characteristics were identified; clones such as 7-3 and 134-1 showed significant binding to mouse HGF as well as human HGF. On the other hand, clones such as 2-3 and 136-3 showed preferential binding only to human HGF with negligible reactivity to mouse HGF. Most of the clones that were reactive to both human and mouse HGF, including clone 7-3, showed significantly reduced reactivity to human HGF in the presence of YYB-101, suggesting that they had the same epitope as YYB-101.

## 3. Discussion

In the development of therapeutic antibodies, it is rare to find antibodies that are cross-reactive to mouse homologues of human epitopes. For example, rituximab (Rituxan, Genentech), alemtuzumab (Lemtrada, Genzyme), trastuzumab (Herceptin, Genentech), cetuximab (Erbitux, Merck), and bevacizumab (Avastin, Genentech) do not bind to rodent homologues of their targets [24]. Therefore, immune-compromised mice engrafted with human tumor cells are often used to test their anti-cancer effects in vivo. In other areas of research, a surrogate antibody that is reactive to a rodent homologue of a human antigen is commonly used. For example, CFA0322, a surrogate antibody that is reactive to mouse complement C5, was employed to test the efficacy of eculizumab (Soliris, Genzyme) [25]. CFA0322 inhibited the cleavage of mouse complement C5, but it is still unclear whether it binds to the homologue of the epitope targeted by eculizumab. Because antibodies with different epitopes on the same antigen behave differently [26], it is ideal for a surrogate antibody to bind to an epitope that is homologous to the epitope targeted by the therapeutic antibody. In a previous study, we developed a pair of anti-complement C5 antibodies that are reactive to homologous epitopes on human and mouse antigens, respectively. Using the antibody that is reactive to mouse complement C5, we showed that binding of the epitope effectively inhibited complement C5 cleavage and choroidal neovascularization [27]. Unfortunately, the development of a surrogate antibody that is reactive to a rodent homologue of a human epitope requires much effort and can be hard to achieve.

We developed a platform technology to monitor changes in *V_H_* genes following immunization, classified the *V_H_* sequences into components based on homology, and selected the components with a high chance of reactivity to a specific antigen. As all the *V_H_* sequences inside a given component shared significant homology, it was reasonable to assume that all the antibodies encoded by those sequences would react to the same epitope. Most of the antibodies encoded by the *V_H_* sequences of component 20 reacted to human HGF and competed with YYB-101 for binding to HGF. Many of them also showed significant reactivity to mouse HGF. The variation in the reactivity of the antibodies to mouse HGF might be due to differences in the artificially-paired *V_K_* genes during the construction of phage-display scFv library or mutations in the *V_H_* genes among individual B cells. Our results show how NGS and in silico analyses of the BCR repertoire can be used to provide a simple, streamlined tool to select antibodies that are reactive to homologous epitopes on human and rodent antigens.

## 4. Materials and Methods

### 4.1. Chicken Immunization

A white leghorn chicken was immunized with 5 µg recombinant human HGF (100-39H; Peprotech, Rocky Hill, NJ, USA). Two booster injections were performed at 2-week intervals. The chicken was sacrificed 1 week after the second booster. Blood samples were collected from wing veins at weeks 0, 2, 4, and 5. The experiment was approved by the Ethics Committee of BioPOA (Seoul) (ethical approval code: BP-2017-032-1, 12 July 2017).

### 4.2. Preparation of cDNA

Peripheral blood mononuclear cells (PBMCs) were isolated from the blood samples using Lymphoprep (#07861; Stemcell Technologies, Vancouver, BC, Canada) according to the manufacturer’s protocol. Total RNA was isolated from the PBMCs using TRI Reagent (Invitrogen, Carlsbad, CA, USA) according to the manufacturer’s protocol. cDNA was synthesized using the Superscript® III First-Strand Synthesis system (Invitrogen) according to the manufacturer’s protocol.

### 4.3. NGS Analysis

NGS analysis was performed as described previously [28]. From the four sets of cDNA constructed using the RNA samples of week 0, week 2, week 4, and week 5, respectively, we amplified *V_H_* genes using a forward primer (5′-ACTCAGCCGTCCTCGGTGTC-3′) and a reverse primer (5′-ACTGACCTAGGACGGTCAGG-3′). We then submitted the amplified *V_H_* cDNA to Celemics, Inc. (Seoul, Republic of Korea) for analysis using the Illumina MiSeq system (Illumina Inc., San Diego, CA, USA). The MiSeq libraries for DNA sequencing were prepared using the SPARK DNA sample prep kit (Enzymatics, Beverly, MA, USA) according to the manufacturer’s protocol. The final libraries were quantified and qualified using D1000 ScreenTape assays with the Agilent 4200 TapeStation according to the manufacturer’s protocol. Long paired-end reads (2 × 300 bp) were generated using the MiSeq reagent kit v3 (Illumina). We uploaded the sequence data to NCBI (SRA accession number: PRJNA494489).

### 4.4. Preprocessing

Preprocessing of the sequence reads comprised three steps: primer annotation, quality filtering, and error correction. The primer annotation was done by a Python script (Python 3.6) using a pattern-matching method as described previously [29]. We filtered the whole sequence data set using FastQC (FastQC v0.11.7, Babraham Bioinformatics) with the ‘q20p100’ option [30]. Finally, error correction was performed using the MiXCR method [15], which corrects errors based on hierarchical clustering.

### 4.5. Sequence Distance Matrix Calculation

We used the edit distance module (https://pypi.org/project/editdistance/) of Python to calculate the LD between every pair of *V_H_* sequences in the whole NGS dataset. The LD between two strings equals the minimum number of single-character edits (including insertions, deletions, and substitutions) required to change one string into the other. 

### 4.6. Component-Based Analysis

Two types of data were used in the component-based analysis: the verified sequence data obtained after preprocessing and the sequence distance matrix. Component-based analysis was performed using R software (R 3.4.3, The R Foundation for Statistical Computing). We used functions embedded in the igraph R package [31] for whole-network construction, component extraction, and network visualization. All statistical analyses were calculated using R scripts, and the results were plotted using the ggplot2 R package (https://github.com/tidyverse/ggplot2).

### 4.7. Phage Display of the Combinatorial scFv Library and Bio-Panning

A phage display of the combinatorial scFv library was constructed using the cDNA obtained from the week 5 blood sample as described previously [32]. Briefly, we used two-step PCR to generate scFv gene sequences, using the cDNA as a template. We digested the scFv gene sequences with *Sfi*I and ligated them into a pComb3XSS phagemid vector. After electroporation of the ligated vector into ER2738 *Escherichia coli* cells, phage was rescued using the M13K07 helper phage. For bio-panning, 1.5 μg recombinant human HGF protein was conjugated to 5 × 10^6^ magnetic beads (Dynabeads M-270 epoxy, Invitrogen) and used the conjugated beads for each round. The library was subjected to four rounds of bio-panning as described previously [33]. The phagemid DNA obtained from overnight cultures of *E. coli* cells after the fourth round of bio-panning was then sent for high throughput retrieval of scFv clones by TrueRepertoire^TM^ analysis [14].

### 4.8. High Throughput Retrieval of scFv-Clones

TrueRepertoire^TM^ is a laser-based, high-throughput platform for the identification and retrieval of clones in an antibody library [14]. About 2000 microcolonies formed on the TR chip. From those, 885 unique variable heavy and light chains were identified, and 306 clones encoding *V_H_* sequences identified in the NGS analysis of the week 5 sample were obtained for phage ELISA.

### 4.9. Phage ELISA

The scFv gene sequences obtained from TrueRepertoire^TM^ were cloned into the pComb3XSS vector. After transfecting them into *E. coli* ER2738 cells, the phages were rescued using the M13KO7 helper phage and subjected them to phage ELISA as described previously [34]. Briefly, microtiter plates (3690; Corning life Sciences, Corning, NY, USA) were coated overnight at 4 °C with 100 ng human HGF (Peprotech) or mouse HGF (Peprotech) in coating buffer (0.1 M sodium bicarbonate, pH 8.6). The wells were blocked with 150 µL 3% (*w*/*v*) bovine serum albumin (BSA) (Thermo Fisher Scientific, IL, USA) dissolved in phosphate buffered saline (PBS) for 1 h at 37 °C. We then incubated the plates sequentially with scFv-displaying phages with or without YYB-101 (100 nM), horseradish peroxidase (HRP)-conjugated anti-M13 monoclonal antibody (GE Healthcare, Pittsburg, PA, USA) in 3% BSA/PBS, and finally 2,2′-azino-bis-3-ethylbenzothiazoline-6-sulfonic acid solution (Pierce, Rockford, IL, USA) with intermittent washing with 0.05% (*v*/*v*) Tween (Sigma-Aldrich, St. Louis, MO, USA) in PBS (PBST). Absorbance was measured at 405 nm with a Multiskan Ascent microplate reader (Labsystems, Helsinki, Finland). Two replicate experiments were conducted for each experimental condition.

## 5. Conclusions

We developed a simple, streamlined process to select antigen-reactive *V_H_* gene components. First, we used NGS to monitor the changes in the BCR repertoire following immunization and boosting. We then clustered the *V_H_* gene sequences into components based on sequence homology. We classified the components by the degree of change in the number of unique *V_H_* sequences and the average read frequency of the *V_H_* sequences inside each component. Because all the *V_H_* sequences within a given component shared significant homology, the antibodies encoded by *V_H_* sequences within the component are likely to react to the same epitope, but they can also have varying characteristics, such as reactivity to human or rodent antigens, depending on the paired light chain or mutations. Using this streamlined process, we successfully generated antibodies that are reactive to both human and mouse HGF and competitive with YYB-101 for binding to human HGF. 

## Figures and Tables

**Figure 1 ijms-20-00417-f001:**
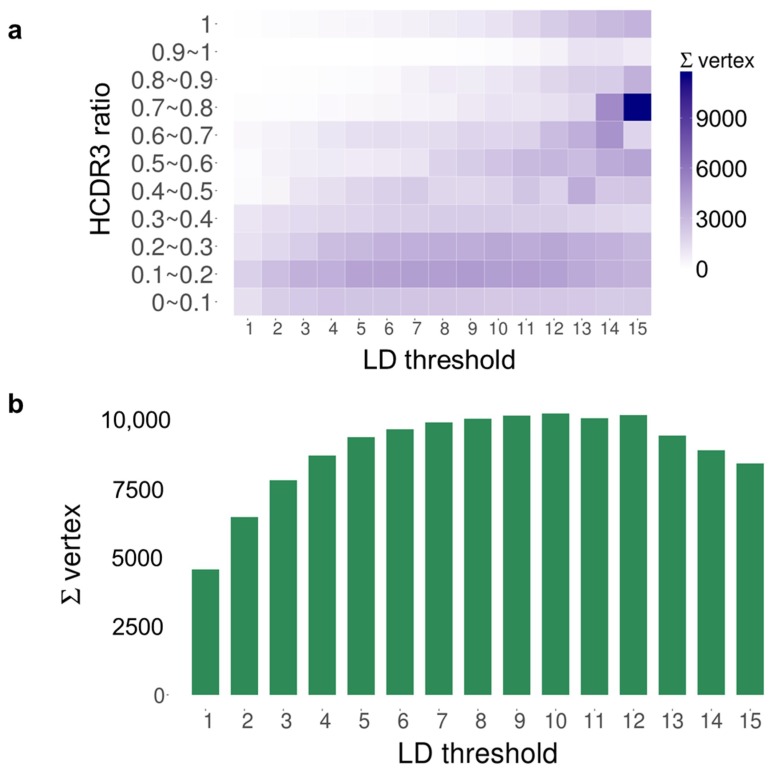
Formation of components depending on the Levenshtein distance (LD) threshold. (**a**) The HCDR3 (CDR3 of *V_H_*) ratio (y-axis) indicates the number of unique complementary determining region 3 (CDR3) amino acid sequences divided by the number of vertices in each component. The heatmap shows the total number of vertices belonging to multiple components within the same range of the HCDR3 ratio. For LD thresholds from 4 to 15, the number of vertices in components with an HCDR3 ratio of 0.1–0.2 or 0.2–0.3 was significantly higher than that in components with an HCDR ratio of 0.3–0.4. Components with fewer than three vertices (*n = 2095*) were excluded. (**b**) Histogram showing the total number of vertices in components with an HCDR3 ratio under 0.3. The maximum number of vertices was achieved with an LD of 12.

**Figure 2 ijms-20-00417-f002:**
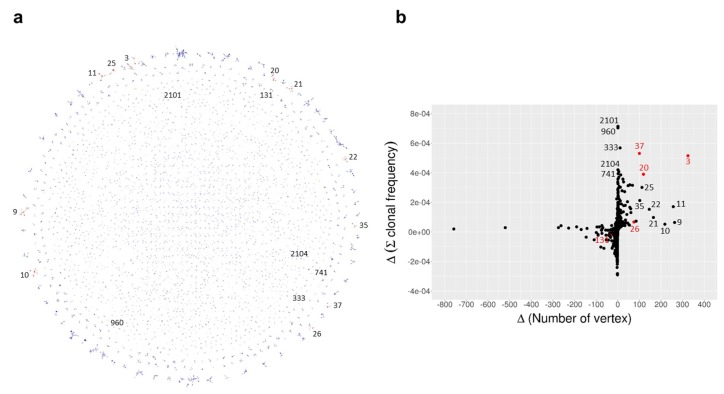
Characteristics of *V_H_* components. (**a**) Network image of *V_H_* components. The vertices in all four sets were connected by finding minimum spanning trees. The diameter of the vertex was drawn depending on the highest clonal frequency among the four sets. Components composed of a single vertex were excluded. Components showing significant increase in the number of vertex and/or clonal frequency from week 0 to week 5 were arbitrarily selected in the scatter plot (**b**) and drawn in brown. In the scatter plot, components confirmed to contain human hepatocyte growth factor (HGF)-reactive clones are labeled in red. All the other components are labeled in black.

**Figure 3 ijms-20-00417-f003:**
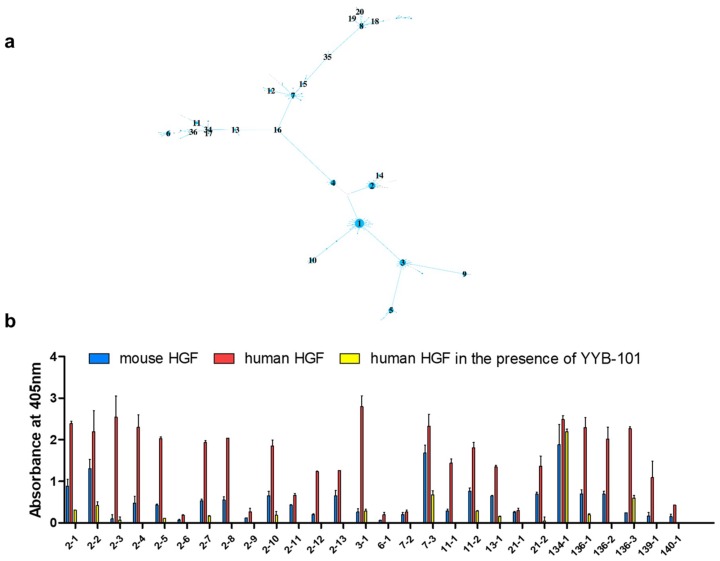
Characteristics of the antibodies encoded by *V_H_* transcripts in component 20. (**a**) Network image of component 20 on week 5 after immunization. The size of each vertex is proportional to its clonal frequency. Vertices are labeled in descending order of clonal frequency. (**b**) Clones from component 20 were rescued and tested for their binding ability to mouse HGF and human HGF in the presence or absence of YYB-101. The absorbance was measured at 405 nm. The first number of the clone name indicates the vertex number, and the second number indicates the *V_K_* gene identifier. Results are shown as the mean ± standard deviation of duplicate experiments.

**Table 1 ijms-20-00417-t001:** Blood sampling, next-generation sequencing (NGS) analysis, and immunization/boosting.

Time Point	Blood Sampling	Immunization	NGS Reads	*V_H_* Gene
Week 0	**O**	**O**	347,331	84,693
Week 2	**O**	**O**	213,534	21,556
Week 4	**O**	**O**	94,925	17,213
Week 5	**O**		95,293	11,309

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
