# Peer review of "Efficient Selection of Antibodies Reactive to Homologous Epitopes on Human and Mouse Hepatocyte Growth Factors by Next-Generation Sequencing-Based Analysis of the B Cell Repertoire"

_ijms, 2019, doi:10.3390/ijms20020417_

Reviewer 1 Report

This paper described the identification of avian antibodies specific to HGF by employing NGS analysis of VH repertoires and biopanning/screening of scFv library constructed from antigen-immunized chicken. The paper provided an interesting method to isolate surrogate antibodies which shared the common recognition epitope with YYB-101 and the authors finally succeeded in the isolation of such surrogate antibodies. The paper is properly organized, and the conclusion is clear. However, the reviewer has the following questions and requirements to improve the manuscript.

1)     In Fig. 3a, only network image of component 20 on week 5 after immunization is displayed. If the authors focus on the changes in antigen-reactive B cells repertoires, they should display network images of component 20 on early week (0, 2 and/or 4 weeks) as well as week 5 after immunization to demonstrate the increase in the number of the vertices.

2)     The journal readers will be probably interested in what kinds of the changes in the sequences were occured and how was the frequencies of typical vertices (in component 20) to trace the clonal expansion or diversion of antigen-stimulated B cells. The reviewer recommend the authors to add such data in the figures..

3)     Why were not scFv clones found from the most major vertex 1 in Fig. 3a and b?

4)     The authors should pick up and describe the clone number of the cross-reactive scFvs toward human and mouse HGF (Fig. 3b) in the Results section.

Author Response

Response to reviewers
-Reviewer 1
Comments and suggestions for authors:

This paper described the identification of avian antibodies specific to HGF by employing NGS analysis of VH repertoires and biopanning/screening of scFv library constructed from antigen-immunized chicken. The paper provided an interesting method to isolate surrogate antibodies which shared the common recognition epitope with YYB-101 and the authors finally succeeded in the isolation of such surrogate antibodies. The paper is properly organized, and the conclusion is clear. However, the reviewer has the following questions and requirements to improve the manuscript.

Comment #1

In Fig. 3a, only network image of component 20 on week 5 after immunization is displayed. If the authors focus on the changes in antigen-reactive B cells repertoires, they should display network images of component 20 on early week (0, 2 and/or 4 weeks) as well as week 5 after immunization to demonstrate the increase in the number of the vertices.

Response to the reviewer’s comment:

We appreciate the reviewer’s helpful suggestion.

We have added new figures (supplementary figures 3a-d) displaying network image of component 20 on early weeks (0, 2 and/or 4 weeks).

Comment #2

The journal readers will be probably interested in what kinds of the changes in the sequences were occured and how was the frequencies of typical vertices (in component 20) to trace the clonal expansion or diversion of antigen-stimulated B cells. The reviewer recommend the authors to add such data in the figures..

Response to the reviewer’s comment:

We appreciate the reviewer’s comment.

We have added a datasheet (supplementary file 1_component20) showing the changes in the sequence and frequency of vertices in component 20.

Comment #3

Why were not scFv clones found from the most major vertex 1 in Fig. 3a and b?

Response to the reviewer’s comment:

We understand the reviewer’s concern. Currently we do not have a clear explanation about this phenomenon. However, one hypothesis is that this clone might have a tendency not to be expressed well in E. coli or displayed efficiently on phage coat surface.

Comment #4

The authors should pick up and describe the clone number of the cross-reactive scFvs toward human and mouse HGF (Fig. 3b) in the Results section.

Response to the reviewer’s comment:

We thank the reviewer for the opportunity to improve our manuscript and have modified the manuscript accordingly (page 6, lines 202-207).

Added and modified sentences:

The VH sequences in component 20 shared 98.3% identity and 98.9% amino acid similarity. We tested the reactivity of the clones to mouse HGF and found that many of the clones in component 20 bound to mouse HGF (Figure 3b). Clones with various binding characteristics were identified; clones such as 7-3 and 134-1 showed significant binding to mouse HGF as well as human HGF. On the other hand, clones such as 2-3 and 136-3 showed preferential binding only to human HGF with negligible reactivity to mouse HGF. Most of the clones that were reactive to both human and mouse HGF, including clone 7-3, showed significantly reduced reactivity to human HGF in the presence of YYB-101, suggesting that they had the same epitope as YYB-101.    

Reviewer 2 Report

in this paper authors reported the generation of surrogate antibodies reactive to mouse HGF, with the future aim to use these molecules for evaluating the efficacy of HGF inhibition on immune-checkpoint inhibitors for cancer therapy.

At first authors performed NGS on chocken PB samples before and after immunization with HGF, identifiying Vh sequences. Then they used the mononuclear cell fraction obtained at week 5 after immunization for phage display and tested  the binding for mouse and human HGF.

i believe that it would improve the readibility if authors explain some points:

1)The last sentence of introduction (lne 83-84) must be eliminated, or used in the discussion section.

2) they performed NGS on 4 PB samples: week 0, and week 2,4,5 after immunization. 

Week 0 and 2 in NGS produced a significantly higher amoun of reads in comparison with week 4 and 5. 

was it wanted?  Can author comment on it?  

3)Authors should explain better how they processed and used data acquired from these 4 points in the method section. Were data merged together or analyzed separately?

Can authors explain better why they sequenced week 0, before immunization? They report to have found 133,312 unique Vh sequence, and since at week 0 they found 84,693 Vh genes, does it means that the great majority of Vh sequence identified originates from week 0? 

4)Figure 2A: explain in the legend the meaning of the numbers. i am not sure to see the "brown labeling" : are numbers or dots brown?

Figure 2B: can authors explain better the figure?  why some some dots has the number in black and other not? 

how can i see the "significant increase in the number of vertices and also in the average clonal frequency from week 0 to week 5"?

5) could authors provide the list of antibodies obtained and tested? or at least some raw data concerning the Vh sequences identified?

6)many methodological informations are provided inside the results section, while methods section is poor

Author Response

-Reviewer 2
Comments and suggestions for authors:

In this paper authors reported the generation of surrogate antibodies reactive to mouse HGF, with the future aim to use these molecules for evaluating the efficacy of HGF inhibition on immune-checkpoint inhibitors for cancer therapy.

At first authors performed NGS on chicken PB samples before and after immunization with HGF, identifiying Vh sequences. Then they used the mononuclear cell fraction obtained at week 5 after immunization for phage display and tested the binding for mouse and human HGF.

I believe that it would improve the readability if authors explain some points:

Comment #1

The last sentence of introduction (line 83-84) must be eliminated, or used in the discussion section.

Response to the reviewer’s comment:

We appreciate the reviewer’s comment and have eliminated the last sentence of introduction.

Deleted sentences:
“In further studies, we plan to verify that the selected antibodies neutralize mouse HGF in vitro and to confirm their effect on the efficacy of immune-checkpoint inhibitors in in vivo“.

Comment #2

They performed NGS on 4 PB samples: week 0, and week 2,4,5 after immunization. Week 0 and 2 in NGS produced a significantly higher amount of reads in comparison with week 4 and 5. was it wanted?  Can author comment on it?

Response to the reviewer’s comment:

We understand the reviewer’s concern. The difference in the sampling read depth was not deliberate. Multiplex sequencing was used to analyze the library. Although we tried to equally distribute the proportion of reads between the libraries, it seems that there was an unequal distribution. However, previous articles reported that sampling depth of 100k reads is sufficient to evaluate a VH library (for example, reference 1 and 2). As we have achieved a similar sampling depth for all the libraries, we proceeded the experiment.

Reference 1: Glanville, J.; D’Angelo, S.; Khan, TA.; Reddy, ST.; Naranjo, L.; Ferrara, F.; Bradbury, AR. Deep sequencing in library selection projects: what insight does it bring?. Curr. Opin. Struct. Biol. 2015, 33, 146.

Reference 2: Galson, JD.; Trück, J.; Fowler, A.; Münz, Z.; Cerundolo, V.; Pollard, AJ.; Lunter, G.; Kelly, DF. In-Depth assessment of within-individual and inter-individual variation in the B cell receptor repertoire. Front. Immunol. 2015, 6, 531

Comment #3

Authors should explain better how they processed and used data acquired from these 4 points in the method section. Were data merged together or analyzed separately? Can authors explain better why they sequenced week 0, before immunization? They report to have found 133,312 unique Vh sequence, and since at week 0 they found 84,693 Vh genes, does it means that the great majority of Vh sequence identified originates from week 0?

Response to the reviewer’s comment:

We thank the reviewer for the opportunity to improve our manuscript. We changed a sentence to provide clear description on the data processing in network analysis (Page 3 Line 115).  The reason why we included VH nucleotide sequences from week 0 in network analysis was to monitor the chronological development of VH sequences. To clarify this, we provided Figure S2 showing the presence and/or absence of each and every vertices of component 20 in network graph in four sets. 

Before change:

To explore the VH transcript sets, we used network analysis tools described previously [16-18].

After change:

To explore these 133,312 VH nucleotide sequences, we used network analysis tools described previously [16-18].

Comment #4

Figure 2A: explain in the legend the meaning of the numbers. i am not sure to see the "brown labeling" : are numbers or dots brown? Figure 2B: can authors explain better the figure?  why some some dots has the number in black and other not? how can i see the "significant increase in the number of vertices and also in the average clonal frequency from week 0 to week 5"?

Response to the reviewer’s comment:

We appreciate the reviewer’s comments.

Following the comment, we included the description about how this number on components were determined in “Results” section (page 5 line 170-172).  In Figure 2A, components marked with number were selected in scatter plot (Figure 2B) as they showed significant increase in in the number of vertex and/or clonal frequency. In these selected components, all the vertices and edges of the selected components were drawn in brown color. We also provided Supplementary Figures 2 in high resolution.  We also modified the legend of Figure 2 following the comments.

Added and modified sentence:

“For the naming of the components, we sorted all components with the number of vertices throughout four time sets in descending order and gave them an identification number in an ascending order”

Comment #5

Could authors provide the list of antibodies obtained and tested? or at least some raw data concerning the Vh sequences identified?

Response to the reviewer’s comment:

We appreciate the reviewer’s comment.

We have added a datasheet (supplementary file 1_component20) showing the changes in the sequence and frequencies of typical vertices in component 20.

Comment #6

Many methodological informations are provided inside the results section, while methods section is poor

Response to the reviewer’s comment:

In writing this manuscript, we decided to allocate a lot of methodological content in “Results” section as  these analytical methods are rather novel and can be considered as results we generated rather than the description of well-known methods.